# Endoscopic Management for Post-Surgical Complications after Resection of Esophageal Cancer

**DOI:** 10.3390/cancers14040980

**Published:** 2022-02-15

**Authors:** Dörte Wichmann, Stefano Fusco, Christoph R. Werner, Sabrina Voesch, Benedikt Duckworth-Mothes, Ulrich Schweizer, Dietmar Stüker, Alfred Königsrainer, Karolin Thiel, Markus Quante

**Affiliations:** 1Department of General, Visceral and Transplantation Surgery, University Hospital Tübingen, Hoppe-Seyler-Str. 3, 72076 Tübingen, Germany; benedikt.mothes@med.uni-tuebingen.de (B.D.-M.); ulrich.schweizer@med.uni-tuebingen.de (U.S.); dietmar.stueker@med.uni-tuebingen.de (D.S.); alfred.koenigsrainer@med.uni-tuebingen.de (A.K.); karolin.thiel@med.uni-tuebingen.de (K.T.); markus.quante@med.uni-tuebingen.de (M.Q.); 2Department of Gastroenterology, Gastrointestinal Oncology, Hepatology, Infectiology, and Geriatrics, University Hospital Tübingen, Otfried-Müller-Str. 10, 72076 Tübingen, Germany; stefano.fusco@med.uni-tuebingen.de (S.F.); christoph.werner@med.uni-tuebingen.de (C.R.W.); sabrina.voesch@med.uni-tuebingen.de (S.V.)

**Keywords:** esophageal cancer, endoscopic complication management, postsurgical complication

## Abstract

**Simple Summary:**

Flexible endoscopy has an important part in the diagnosis and treatment of postoperative complications after oncologically intended esophagectomy. Endoscopy offers the possibility of effective therapy with minimal invasiveness at the same time, and the use of endoscopic therapy procedures can avoid re-operations. In this review we present the advantages of endoscopic treatment opportunities during the last 20 years regarding patients’ treatment after esophageal cancer resection. According to prevalence and clinical relevance, four relevant postoperative complications were identified and their endoscopic treatment procedures discussed. All endoscopic therapy procedures for anastomotic bleeding, anastomotic insufficiencies, anastomotic stenosis and postoperative delayed gastric emptying are presented, including innovative developments.

**Abstract:**

Background: Esophageal cancer (EC) is the sixth-leading cause of cancer-related deaths in the world. Esophagectomy is the most effective treatment for patients without invasion of adjacent organs or distant metastasis. Complications and relevant problems may occur in the early post-operative course or in a delayed fashion. Here, innovative endoscopic techniques for the treatment of postsurgical problems were developed during the past 20 years. Methods: Endoscopic treatment strategies for the following postoperative complications are presented: anastomotic bleeding, anastomotic insufficiency, delayed gastric passage and anastomotic stenosis. Based on a literature review covering the last two decades, therapeutic procedures are presented and analyzed. Results: Addressing the four complications mentioned, clipping, stenting, injection therapy, dilatation, and negative pressure therapy are successfully utilized as endoscopic treatment techniques today. Conclusion: Endoscopic treatment plays a major role in both early-postoperative and long-term aftercare. During the past 20 years, essential therapeutic measures have been established. A continuous development of these techniques in the field of endoscopy can be expected.

## 1. Introduction

Esophageal cancer ranks seventh in terms of incidence and sixth in overall mortality of cancer-related deaths in the world, with relevant differences in regional frequency [1]. The incidence of esophageal cancer (EC), especially adenocarcinomas of the gastroesophageal junction (AEG), has been rising rapidly during the last decades.

Surgical resection remains the “gold standard” of curative treatment for esophageal cancer [2]. According to the size and location of the tumor, the following different surgical methods and treatment strategies can be utilized. Standard esophageal resection procedures are:McKeown technique = right thoracotomy followed by laparotomy, gastric tube formation for conduit and neck incision with cervical anastomosis;Ivor Lewis esophagectomy = right thoracotomy and laparotomy with gastric tube formation for conduit;and Transhiatal approach = oncological gastrectomy with extended distal esophageal resection.

All types of esophageal surgery are increasingly performed as minimally invasive procedures [2,3,4,5]. Esophagectomy remains an example of complex major surgery associated with a significant risk of major morbidity and a substantial impact on health-related quality of life. Mortality after esophageal resection has decreased significantly from over 30% in the 1970s to below 5% in specialized centers today, but morbidity rates remain high with up to 50% reported [6]. 

Endoscopic procedures can avoid surgery for early stages of EC [7,8,9]. Endoscopic resection techniques, endoscopic mucosal resection (EMR) or endoscopic submucosal dissection (ESD), and submucosal tunneling endoscopic resection (STER) or newly endoscopic full-thickness resection (EFTR), can completely remove early lesions. 

In addition to its role as a diagnostic and primary endoscopic therapeutic measure, endoscopy plays also a relevant role in the treatment of postoperative complications after EC resection [10]. Postoperative complications that may require endoscopic therapy are as follows: anastomotic hemorrhage, anastomotic insufficiency, delayed gastric emptying, and anastomotic stenosis. Anastomotic hemorrhage is an acute postoperative complication. In order not to harm the anastomosis itself and not to restrict its blood flow, the available hemostasis techniques should be well known. Anastomotic insufficiencies represent the most serious type of complications, since they are associated with increased morbidity and mortality. Delayed gastric emptying and anastomosis stenoses occur in a delayed time when the patient re-starts to eat solid food again. Then, relevant problems may arise due to regurgitations and restricted nutrition. Several innovative techniques have been developed in recent years to treat patients after surgical esophagectomy with a minimally invasive, endoscopic approach to optimize the perioperative care of patients and to treat complications effectively, thus finally minimizing morbidity. 

## 2. Materials and Methods

### 2.1. Evaluation of the Main Topics

The selection of the main topics for the review is based on their prevalence and clinical relevance: anastomotic hemorrhage, anastomotic insufficiency, conduit ischemia, delayed gastric emptying, and anastomotic stenosis.

### 2.2. Literature Search

A literature search according to the revised PRISMA guidelines 2020 for each of the four postsurgical complications following EC resection was performed by screening the electronic databases MEDLINE (via Ovid SP), EMBASE (via DIMDI), Web of Science, and the Cochrane Central Register of Controlled Trials (CENTRAL) covering the period from January 2000 until June 2021. In sum, 4871 studies were analyzed for this review. See the review process in the flowchart of Figure 1. Case reports with at least three patients were included for this review. The search strategies for the databases were adapted to the specific vocabulary of each database. Moreover, the references of the articles included were then manually screened to identify additional relevant studies. For each topic, two authors independently reviewed the title and abstract of all records defined by the systematic literature search as well as the full texts of all articles assessed for eligibility.

## 3. Results

### 3.1. Anastomotic Hemorrhage

Anastomotic hemorrhage following EC resections is rare, but if it occurs, immediate treatment is needed [11]. Rate of bleeding is reported in up to 2% of patients following esophagectomy or gastrectomy for EC [12]. 

Endoscopic treatment of hemorrhage in the gastrointestinal (GI) tract is based on three different procedures: elevated pressure to the submucosal layer due to mechanical measures, thermic hemostasis and application of absorbent substances. For anastomotic hemorrhage all these measures are available, too [13]. The choice of hemostasis technique depends on the preferences of the endoscopist and the age of the anastomosis. In the subsequent section, the corresponding measures including innovations and the existing literature of the last 20 years are presented. Clinical papers reporting exclusively on hemostasis in the anastomotic area after esophageal resection are rare, so additional articles were selected for the presentation of the substances and techniques used in the context of anastomotic hemorrhage. Yet, no randomized controlled trial (RCT) has compared the therapeutic efficacies of the various modalities for postoperative hemorrhage. Figure 2 shows different endoscopic hemostasis techniques using the appropriate tools in the endoscope working channel or placed on the distal end.

#### 3.1.1. Mechanical Pressure Build-Up in the Area of the Submucosal Layer

##### Application of Through-the-Scope-Clips

The Hemoclip or Endoclip as a mechanical method of hemostasis was first introduced in 1975 [11,14,15]. It is widely used because of its simplicity, low cost, easy availability, repetition, minimal damage to the localized field, and reduced risk of adverse effects for the treatment of nonvariceal bleeding in the upper GI tract [11]. The use of Hemoclips offers an application via the working channel with a permanent view to the bleeding source. Hemoclips are suitable for anastomotic bleeding and do not injure the surrounding tissue [16,17]. 

##### Application of Over-the-Scope-Clips (OTSC)

The “bear claw” clip, or OTSC, is an excellent tool for primary and secondary bleeding situations in the GI tract [13,18]. However, no literature on the use of OTSC in the context of anastomotic hemorrhage after EC resections is currently available. The OTSC provides a high closure power, thus, an impairment of anastomotic blood circulation due to the clip may be possible. Tontini et al. reported a case of successful hemostasis using the OTSC for an anastomotic bleeding situation after Bilroth I resection [19]. Endoscopic devices such as the Twin-Grasper can enable safe and precise application of the clip. The first description of OTSC in clinical use was reported by Kirschniak et al. in 2007 [20]. Meanwhile, OTSC application has become one of the standard procedures for bleeding events in the gastrointestinal tract [13,21].

#### 3.1.2. Injection Therapy

Hemostasis can also be achieved with endoscopic injection therapy using epinephrine or fibrin glue. The mechanism of hemostasis is based on pressure elevation in the submucosal layer. Advantages of this simple technique are wide availability and relatively low costs [22].

#### 3.1.3. Thermic Hemostasis

Thermal contact probes have been the mainstay of endoscopic hemostasis since the 1970s. Contact probes provide hemostasis through two distinct mechanisms: (1) tamponade of a blood vessel to stop bleeding and interrupt underlying blood flow and (2) application of thermal energy to seal the underlying vessel (coaptive coagulation). Perforation is a risk of using a thermal probe [22]. Possible probes of the thermic hemostasis techniques are the dry monopolar electrode, the liquid electrode, the bipolar electrode, the heater probe, the coagrasper hemostatic forceps or hot biopsy forceps, and the Argon-plasma-coagulation catheter. All these devices are usually not utilized in anastomotic bleeding situations in fresh esophagogastrostomies or jejunostomies. 

#### 3.1.4. Application of Absorbent Substances

Application of topical hemostatic powder or gel is a new endoscopic treatment concept. Here, different powders and one gel product are available.

##### Mineral-Based Absorbents

The mineral-based Hemospray (Cook Medical, Winston-Salem, NC, USA) is a biologically inert powder. When Hemospray comes in contact with fluids, it creates a mechanical barrier [13]. For anastomotic bleeding situations after EC resections, only a few clinical reports exist. Granata et al. reported on the successful use of Hemospray in a patient with a bleeding complication after gastroenteric anastomosis [23].

##### Polysaccharide-Based Absorbents

The polysaccharide-based powder (EndoClot, EndoClot Plus Inc, Santa Clara, CA, USA) is derived from purified plant starch. The application mode is similar to the Hemospray, where the pressure for application of the powder is generated by an external compressor. A preliminary study on hemostasis for post-EMR lesions in anti-thrombotic treated pigs showed encouraging experimental results [24]. In detail, comparison of Hemospray and Endoclot showed effective hemostasis and no differences in short- or long-term success or rebleeding situations [24]. Yet, there are no clinical reports on the use of Endoclot in postoperative bleeding situations after EC resections available.

##### Hemostatic Gel

PuraStat (3-D Matrix Europe SAS, France) is a novel hemostatic agent, which builds a transparent, stable coating on the mucosa that promotes an immediate and durable hemostatic effect, and it is ready to use. The efficacy of the gel has so far been demonstrated in post-interventional bleeding with a success rate of 90%. [25,26,27,28] For treatment of postoperative bleeding situations after EC resections, no data are available at the moment.

### 3.2. Anastomotic Insufficiency

The reported mortality linked to anastomotic insufficiency (AI) ranges between 2% and 12% [29,30]. Depending on the surgical method and height of the anastomosis, the incidence of AI varies considerably and ranges from 5% to 40% following resection for EC [31]. For cervical AI, external cervical drainage is the best therapeutic method. Clinical findings of intrathoracic AI are fever, pain, respiratory deficiency, conspicuous secretions via thoracic drainage, and elevated laboratory parameters for inflammation, especially high C-reactive protein levels [32,33,34]. If anastomotic fistula or AI occurs, early diagnosis is critical for the patient. The current “gold standard” diagnostic tool is the non-invasive CT examination [33]. Endoscopy is another relevant diagnostic tool. Although early insufficiencies may not be endoscopically detected in all cases, a subsequent CT scan performed after endoscopy may detect air bubbles around the anastomosis, thus proving AI.

For the endoscopic treatment of AI, different innovations were established over the last years. Stent placement and debridement as well as endoscopic negative pressure therapy are new milestones in the management of these postoperative complications. A new treatment option is the VACStent, a combination of stenting and endoscopic negative pressure therapy, which is also used for complications after esophageal resections [35]. 

#### 3.2.1. Stent Placement

As early as 1985, Ravo et al. described an attempt to treat esophageal leaks using a surgically placed intraluminal bypass tube [36]. Similarly, in 1996, Segalin et al. described the temporary sealing of recurrent postoperative leakage using a Wilson-Cook esophageal prosthesis in two patients [37]. The therapeutic principle of stent therapy is based on covering the intestinal defect. Paraesophageal retention formations and fluid collections must then be subsequently drained via additional drainage that has to be surgically placed or radiologically guided.

Two kinds of stents can be used to treat AI: self-expandable plastic and metal. Nowadays, metal stents are almost exclusively used because plastic stents resulted in increased morbidity due to higher migration rates. In addition to the material of the self-expandable metal stent (SEMS), it is necessary to differentiate between fully- and partially covered stents. Fully covered (FC) stents also have an increased risk of dislocation but can be easily removed after the relatively short treatment period for anastomotic insufficiencies. Here, time of stent treatment is reported to range from 2 weeks to 12 months [38,39,40].

Table 1 is providing the publications on stent therapy of leakages after esophagectomy. Overall healing rate of stent-based therapy after surgical resections for EC is 83.47%, with a mortality rate of 13.66%. Stent-based treatment of AI the most common endoscopic technique worldwide. 

#### 3.2.2. Closure of Anastomotic Fistula Using OTSC

In a few articles, the use of OTSC for closure of a chronic esophagojejunal fistula has been described [56,57]. Success rates are depending on local and systemic inflammatory reaction as well as the time point of diagnosis and therapy of the insufficiency. Taken together, there is no general recommendation for primary utilization of this technique for this indication.

#### 3.2.3. Endoscopic Internal Drainage

The use of a double-pigtail endoscopically placed into an anastomotic insufficiency after EC resection is a technique to drain the para-esophageal cave. This endoscopic therapeutic procedure has been rarely reported for AI following EC resections. However, this technique is well described for leakage after bariatric surgery [58]. 

#### 3.2.4. Transnasal Inner Drainage

Kosumi et al. first reported about the use of transnasal inner drainage to manage anastomotic leaks after successful EC resections in five patients [59].

#### 3.2.5. Endoscopic Negative Pressure Therapy

Endoscopic negative pressure therapy (ENPT) is also known as endoscopic vacuum therapy (EVT) or as endoscopic vacuum-assisted closure (EVAC). The underlying therapeutic principle is application of a continuous negative pressure acting on the tissue due to open-pore suction devices (OPSD) consisting of a foam element and a drain. The OPSD is placed on the level of the leak endoluminal or direct in a perforation cave, also called an intracavitary. Intracavitary therapy is used for large, contaminated wound cavities with a broad entrance. The endoluminal position of the OPSD is used for small defects and not for big wound cavities. The drain is then connected to a negative pressure source such as an an electric pump, outside of the patient. 

Since 2008, numerous articles have been published on the use of ENPT in patients after EC resection [60,61,62,63,64,65,66,67]. Most articles are case reports or case series, summarizing a variety of causes for perforations. Taken together, clinical results of ENPT for the treatment of AI after EC resection are good, with frequent endoscopic follow-up of patients and change intervals ranging from 3 to a maximum of 10 days. Especially for acute therapy in mediastinitis with incipient sepsis, ENPT can achieve amazing therapeutic success [68,69,70]. For the therapy of AI after EC resections, inpatient treatment of patients is necessary. Parallel enteral feeding is only possible with a nasojejunal tube in place.

In Table 2, all publications about ENPT in patients with AI after EC resections are listed. Overall healing rate for patients with AI can be summarized at 85.93%, while the mortality rate in all treated patients is 11.34%.

#### 3.2.6. VACStent

The VACStent is a new device which combines the advances of stent application and ENPT with the opportunity of early enteral feeding due to a fully covered stent (Figure 3). Negative pressure is applied to an outer layer of a polyurethane sponge cylinder outside the stent due to a drain. The first case report about the use of a VACStent was published in 2019 by Chon et al. The authors reported about the successful treatment of a patient with AI following oncological gastrectomy for gastric cancer [86]. The first case report about the use of a VACStent for the treatment of AI following EC resection was first described by Lange et al. in June 2021 [35]. 

#### 3.2.7. Preventive Endoscopic Negative Pressure Therapy 

Based on the encouraging clinical results of ENPT for AI following EC, the concept of preemptive ENPT was established (so called pEVT). In 2017, Neumann et al. had already reported on the use of preemptive ENPT in eight patients without manifest AI but with localized anastomotic ischemia [87]. In two patients, a small, localized AI developed during the early clinical course, which was resolved by continuing the ENPT. In the other six patients, complete mucosal recovery was achieved by pENPT. In addition, Gubler et al. reported in 2019 about the successful use of preemptive ENPT in 19 patients, immediately after completion of EC surgery. The PU sponge was then removed after 4 to 6 days. Only one patient had to undergo surgical revision for a secondary AI [88]. 

Of note, a prospective, multicenter, parallel-group RCT started in 2020 as the “preSPONGE Trial” in Switzerland for adult high-risk patients undergoing minimally invasive transthoracic esophagectomy. PU-Sponge placement and start of the ENPT will be established intraoperatively in the intervention group. The investigators are planning to enroll a total of 100 patients (approximately 50 per group) to reach sufficient statistical power [89].

### 3.3. Postoperatively Delayed Gastric Emptying

Postoperatively delayed gastric emptying (PDGE) affects the gastric transport of solid and/or liquid foods [90]. Severity of PDGE can be assessed by delayed gastric emptying scintigraphy (GES) and with the new measurement probe Endoflip™ [91,92]. The Endoflip™ is a distensibility measurement probe via a balloon catheter. Endoflip™ is the acronym for endoscopic Functional Lumen-Imaging Probe. It has been primarily used to evaluate the lower esophageal sphincter in patients suffering from achalasia or gastroesophageal reflux disease [93]. The Endoflip™ measurement offers a precise diagnostic tool and therapy-success review method [94]. 

PDGE after esophagectomy with gastric replacement can result in significant postoperative symptoms such as sensation of early fullness during eating, nausea, dysphagia, chest tightness, cough-induced vomiting, and aspiration. Konradsson et al. report the rate of PDGE with up to 20% [90]. 

#### 3.3.1. Balloon-Dilatation

The use of balloon dilatation is currently studied by different working groups. A benefit for PDGE has been documented with repetitive dilatations up to 35 mm diameter [95]. According to the ESGE statement, a recommendation was given for hydraulic dilation with through-the-scope balloons and pneumatic dilation for at least 1–2 min [96]. 

#### 3.3.2. Injection Therapy with Botulinum Toxin

Endoscopic injection of botulinum toxin should decrease the pyloric tone, thereby improving gastric emptying. Although botulinum toxin injection showed promising results [97,98], two recent RCTs did not show any benefit when compared with a placebo (saline injection) [99]. Thus, ESGE recommendations do not support the use of botulinum toxin injection in non-selected patients with gastroparesis [96]. 

#### 3.3.3. Pyloric Stenting

This technique is associated with a high rate of stent migration [100]. Thus, the ESGE does not recommend the use of transpyloric stenting for the treatment of gastroparesis [96]. 

#### 3.3.4. Gastric Peroral Endscopic Myotomie (G-POEM)

For the G-POEM, a submucosal tunnel is established starting at the antrum, then the pyloric musculature is accessed, and a selective myotomy of the pyloric circular muscle is performed [101]. Currently, available studies and reviews on G-POEM report on the effectivity and safety of this procedure in patients with variable causes of the delayed gastric emptying. A total of seven studies were evaluated reporting on post-surgical patients [102,103,104,105,106,107,108]. However, the type or procedure of previous surgery was not provided in any of those articles. Therefore, ESGE recommends consideration of G-POEM only in carefully selected patients due to overall limited data on effectiveness, safety, and durability [96].

### 3.4. Anastomotic Stenosis/Stricture 

Stenoses can cause dysphagia, odynophagia, and aspiration [109]. Inadequate dietary intake and malnutrition may be results of anastomotic stenosis. Patients with clinical symptoms suggestive of postoperative stricture should undergo endoscopy for further diagnosis. The mainly benign strictures can be effectively managed by endoscopic dilatation and bougie dilatation. Helminen et al. reported about endoscopic treatment for stenosis in patients after surgery with minimally or open approach. The overall incidence of strictures was 16.7% in this analysis [110]. Ahmed et al. reported a rate of 24.5% of anastomotic strictures or stenoses in patients following esophageal resection and reconstruction in a prospective trial [109]. Hanyu and colleagues reported about a stenosis rate of 41% in a retrospective analysis [111]. Finally, Ahmed et al. reported on two dilatations (range number of interventions 1–18) for clinical successful treatment [109]. A new single-use tool for bouginage under endoscopic, visual control is the BougieCap (OVESCO, Tübingen, Germany). 

## 4. Discussion

In this review, we present advantages in endoscopic diagnostic and treatment opportunities during the last 20 years for patients after surgical EC resection. The flexible endoscopy plays an important role for the management of acute and delayed complications in patients after EC resections offering numerous effective treatment strategies for anastomotic bleeding, anastomotic insufficiency, PDGE, and anastomotic strictures. Numerous therapeutic strategies have evolved over time, and new methods and devices for endoscopic diagnosis and therapy still enter the clinical stage every year. In several cases, technical developments were initially implemented for other indications and then adapted for successful utilization in the context of complications after EC resections. Here, endoscopic negative pressure therapy represents a prominent example.

In addition to technical developments in endoscopic diagnostics and therapy, there were also substantial developments in surgical resection techniques and perioperative patient care. Improved perioperative management, early postoperative enteral nutrition, pulmonary physiotherapy, prevention of hypoxemia and hypotension, laparoscopic and thoracoscopic resection techniques, and real-time intraoperative quantitative fluorescent-guided perfusion assessment during resections are important measures resulting in reduces incidence of postoperative complications [6,112]. 

This review is reporting on endoscopic treatment of four complications after EC resection: anastomotic hemorrhage, anastomotic insufficiency, anastomotic stenosis, and PDGE. Here, the first two conditions represent acute, early-postoperative complications. These complications usually occur immediately after surgery or within approximately 1–5 days postoperatively [33]. In contrast, the latter two complications occur with a latency of up to 3 or 6 months after surgical resection. An early flexible endoscopy for fresh anastomoses is frequently performed for follow-up nowadays. Here, it is a critical and important fact that only low pressure is applied to the endoscope in the anastomosis area, and that the examination is performed with carbon dioxide [113].

Hemostasis techniques currently use TTSCs that have been available since the 1980s. These devices have been undergoing further development and refinement, so current TTSC models can also rotate and re-open. Other devices have been newly designed and developed, such as hemostasis powders and the OTSC [13]. The overall success rates of the new hemostasis techniques are good. However, high quality data on therapeutic options for the anastomotic bleeding situation after EC resection is currently still lacking.

For the treatment of anastomotic insufficiencies, impressive clinical success rates could be achieved by utilizing the therapy methods presented in this review. Overall mortality rate of patients with anastomotic insufficiencies steadily decreased from 35.7% [114] at the beginning of 2000 to 11.8% [115] nowadays. This reduction in mortality has been realized through many different achievements in the complex interdisciplinary care of these critically ill patients [116], with flexible endoscopy playing a central role. If anastomotic insufficiency is detected, an immediate intervention with adequate drainage and minimization of further contamination is essential. The use of fully covered metal stents in addition to CT-guided or laparoscopic placed drains or the use of ENPT can significantly reduce patients´ mortality and morbidity. The prevention of anastomotic insufficiencies after EC resection in high-risk patients using pENPT is a new, promising approach [88,89]. Future publications will help to define the role of preventive negative pressure therapy in the anastomosis area. 

The long-term complications of anastomotic stenosis and PDGE are primarily treated mechanically. The incidence of anastomotic stenoses could also be reduced without a reduction of blood flow in the anastomotic region by preventive measures [31,90]. For anastomotic stenosis, balloon dilatation is currently available as the “gold standard” treatment option.

In the context of PDGE, the introduction of impedance planimetry (Endoflip™) may result in significant advances for the therapy of this complication. This system allows accurate diagnosis and verification of therapeutic success, can be used repeatedly, and may also help to identify patients at risk for the development of PDGE later on [93,94]. 

## 5. Conclusions

Endoscopic treatment plays a major role in both immediate and delayed aftercare. During the past 20 years, essential therapeutic measures have been established. A continuous development of these achievements in the field of endoscopy can be expected. 

## Figures and Tables

**Figure 1 cancers-14-00980-f001:**
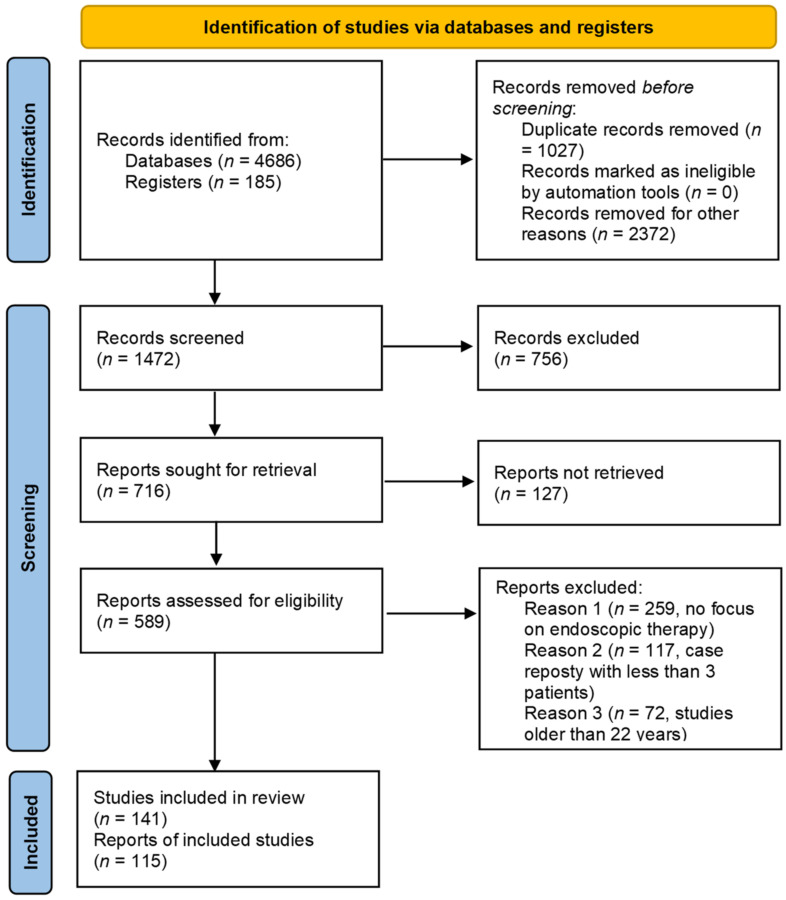
PRISMA 2020 flow diagram for the review process.

**Figure 2 cancers-14-00980-f002:**
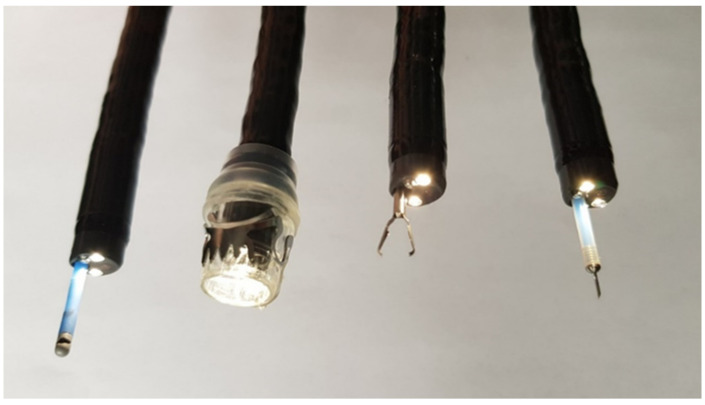
A selection of instruments for endoscopic haemostasis (from left to right): APC lateral probe, OTSC, TTSC haemostasis clip, injection needle. With kind permission from Dr. J. Fundel and U. Schweizer.

**Figure 3 cancers-14-00980-f003:**
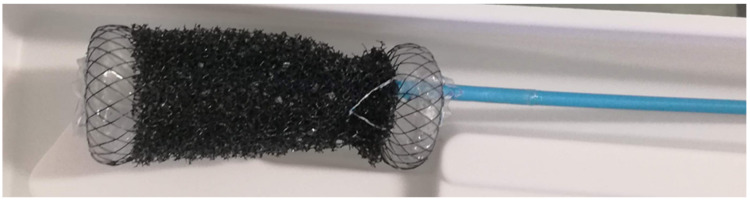
New combination device VACStent (Möller Medical GmbH, Fulda, Germany); the blue drain is connected to an electric vacuum pump. The device could be in place for 5 days.

**Table 1 cancers-14-00980-t001:** Publications with more than five patients included with stent-based therapy in patients with AI after EC resections.

Reference	Year	Study Type	Number of All Patients	Number of Patients after EC Resection	Success Rate in Patients (All Patients)	Mortality Rate (All Patients)
Roy-Choudhury et al. [41]	2001	Case series	14	14	92.86%	7.14%
Hünerbein et al. [42]	2004	Case series	9	9	100%	0%
Langer et al. [43]	2005	Case series	24	24	n.n.	25%
Schubert et al. [44]	2005	Case series	12	12	91.67%	0%
Radecke et al. [45]	2006	Case series	9	5	66.67%	44.44%
Han et al. [46]	2006	Case series	8	8	100%	12.5%
Freeman et al. [47]	2007	Case series	21	21	95.24%	4.76%
Kim et al. [11]	2008	Case series	17	7	76.47%	5.88%
Tuebergen et al. [48]	2008	Case series	32	24	78.13%	15.63%
Babor et al. [49]	2009	Case series	5	4	100%	0%
Salminen et al. [50]	2009	Case series	10	2	80%	50%
Van Heel et al. [51]	2010	Case series	33	1	69.69%	21.21%
D’Cunha et al. [52]	2011	Case series	37	22	59.46%	13.51%
Cerna et al. [53]	2011	Case series	5	4	80%	0%
Schweigert et al. [54]	2011	Case series	22	22	77%	23%
Smith et al. [55]	2020	Case series	11	11	90.9%	9.1%
Total			269	190	78.63%	14.51%

n.n. = not named.

**Table 2 cancers-14-00980-t002:** Publications with more than five patients on ENPT in patients with AI after EC resections.

Reference	Year	Study Type	Number of All Patients	Number of Patients after EC Resection	Success Rate in Patients after EC Resections	Mortality Rate (All Patients)
Weidenhagen et al. [71]	2010	Case series	6	6	100%	16.67%
Brangewitz et al. [72]	2013	Case series	32	28	n.n.	15.62%
Schniewind et al. [73]	2013	Case series	17	14	n.n	11.76%
Schorsch et al. [74]	2014	Case series	35	21	95.24%	5.17%
Mennigen et al. [66]	2015	Case series	22	19	86.36%	13.64%
Möschler et al. [75]	2015	Case series	10	5	40%	n.n.
Kuehn et al. [76]	2016	Case series	21	11	81.82%	4.76%
Laukötter et al. [68]	2017	Case series	52	39	89.74%	9.61%
Mencio et al. [77]	2018	Case series	15	2	100%	0%
Bludau et al. [78]	2018	Case series	77	51	80.39%	11.69%
Pournaras et al. [79]	2018	Prospective cohort study	21	7	100%	14.29%
Noh et al. [67]	2018	Case series	12	11	91.67%	8.33%
Ooi et al. [80]	2018	Case series	10	3	66.67%	30%
Valli et al. [81]	2018	Case series	12	7	71.42%	0%
Min et al. [82]	2019	Case series	20	13	n.n.	5%
Leeds et al. [83]	2019	Case series	19	2	100%	0%
Jeon et al. [65]	2020	Case series	22	22	86.36%	0%
Jung et al. [84]	2021	Case series	30	23	87%	6.67%
Zhang et al. [85]	2021	Case series	55	55	87.5%	3.9%
Hayami et al. [64]	2021	Case series	23	23	82.6%	13.04%
Total			489	350	85.10%	11.34%

n.n.= not named.

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
