# Peer review of "Endoscopic Management for Post-Surgical Complications after Resection of Esophageal Cancer"

_cancers, 2022, doi:10.3390/cancers14040980_

Round 1
Reviewer 1 Report
Interesting summary of endoscopic treatment of postoperative complication after esophageal resection.
I would recommend minor revisions:
- Please give the number of analyzed studies in the review
- Please provide a flow chart of the review process
68-71: Consider evaluation of main topics out of prevalence and not out of clinical practice. If this is the case, please mention it.
72-82: Was literature search done according to PRISMA-guidelines? If yes, please mention.
137: Think about mentioning hemostatic gel.
208-209: Please state in which case EVT should be placed intraluminal (no relevant extraluminal fluid collection or no big wound cavity) and when not.
287: Data for bouginage for strictures?
Please proof-read your work for minor grammatic corrections.
Author Response
Dear reviewer, I would like to thank you for reviewing the manuscript, for the supportive comments and valuable advices. Your mentioned points and remarks were very helpful. See the point-by-point answer below:
- Please give the number of analyzed studies in the review: The number of 4871 studies is documented now in line 78
- Please provide a flow chart of the review process: The flowchart is implemented
- 68-71: Consider evaluation of main topics out of prevalence and not out of clinical practice. If this is the case, please mention it. : Thanks for this remark. The sentence:
"The selection of the main topics for the review is based on their prevalence and clinical relevance: anastomotic hemorrhage, anastomotic insufficiency, conduit ischemia, delayed gastric emptying, and anastomotic stenosis." is inserted in line 70-72
- 72-82: Was literature search done according to PRISMA-guidelines? If yes, please mention. Thanks for this valuable remark. We inserted the infomration and the flowchart
- 137: Think about mentioning hemostatic gel. This was a very heplful comment. We inserted infomrations about the hematsatic gel in line 150 and 166-171.
- 208-209: Please state in which case EVT should be placed intraluminal (no relevant extraluminal fluid collection or no big wound cavity) and when not. We inserted the sentence: Intracavitary therapy is used for large, contaminated wound cavities with a broad entrance. The endoluminal position of the OPSD is used for small defects and no big wound cavities. in line 231-233.
- 287: Data for bouginage for strictures? Thank for this remark, we inserted infomrations about bougienage and the new tool BougieCap in this paragraph in line 316/317 and 323-325.
- Please proof-read your work for minor grammatic corrections: the manuscript is total revised, thanks for this remark.
Thanks again! Your Dörte Wichmann
Reviewer 2 Report
First I want to thank for the opportunity to review the manuscript titled "Endoscopic Management for Post-Surgical Complications after Resection of Esophageal Cancer" submitted by Wichmann and colleagues.
I have read this comprehensive review with great interest because I have been working as a thoracic and general surgeon for nearly 20 years and I´m very, very familiar with these different postoperative complications after esophagectomy and reconstrcution by gastric-pull up.
The authors have adressed the most frequent types of complications in a very satisfying way which enables the experienced and interested reader/ surgeon to get an update on the current therapeutic treatment options in a very short time.
No further queries.
Good luck for the authors!
Author Response
Dear reviewer, thank you very much for your positive assessment of the manuscript. We also had a lot of enjoyment working on it and gained interesting insights. Thanks again, Your Dörte Wichmann